

# Factors associated with intention to be vaccinated with the COVID-19 booster dose: a cross-sectional study in Peru

Rodrigo Camacho-Neciosup[1,2], Ericka N. Balcazar-Huaman[1,2], Margarita L. Alvarez-Vilchez[3,4], Janith P. De la Cruz-Galán[5,6], Yubely Gálvez-Guadalupe[7,8], Edwin D. Garcia-Muñoz[9], Greysi Cerron-Daga[10], Virgilio E. Failoc-Rojas[11] and Mario J. Valladares-Garrido[12]

[1] Sociedad Científica de Estudiantes de Medicina, Universidad Nacional Pedro Ruiz Gallo, Lambayeque, Peru
[2] Facultad de Medicina Humana, Universidad Nacional Pedro Ruiz Gallo, Lambayeque, Peru
[3] Sociedad Científica de Estudiantes de Medicina, Universidad Peruana Los Andes, Huancayo, Peru
[4] Facultad Medicina Humana, Universidad Peruana Los Andes, Huancayo, Peru
[5] Sociedad Científica de Estudiantes de Medicina Veritas, Universidad de San Martín de Porres, Lambayeque, Peru
[6] Facultad de Medicina Humana, Universidad de San Martín de Porres, Chiclayo, Peru
[7] Sociedad Científica de Estudiantes de Medicina, Universidad Privada Antenor Orrego, Trujillo, Peru
[8] Universidad Privada Antenor Orrego, Trujillo, Peru
[9] Escuela de Medicina, Universidad Nacional Toribio Rodríguez de Mendoza de Amazonas, Chachapoyas, Peru
[10] Universidad Nacional Daniel Alcides Carrión, Pasco, Peru
[11] Universidad San Ignacio de Loyola, Lima, Peru
[12] Universidad Cesar Vallejo, Piura, Peru

Corresponding authors
Virgilio E. Failoc-Rojas,
virgiliofr@gmail.com
Mario J. Valladares-Garrido,
Josvg44@gmail.com

## ABSTRACT

**Introduction.** The pandemic of COVID-19 continues to impact people worldwide, with more than 755 million confirmed cases and more than 6.8 million reported deaths. Although two types of treatment, antiviral and immunomodulatory therapy, have been approved to date, vaccination has been the best method to control the spread of the disease.

**Objective.** To explore factors associated with the intention to be vaccinated with the COVID-19 booster dose in Peru.

**Material and Methods.** Cross-sectional study, using virtual and physical surveys of adults with two or more doses of COVID-19 vaccine, where the dependent variable was the intention to be vaccinated (IBV) with the booster dose. We calculated prevalence ratios with 95% confidence intervals, using generalized linear models of the Poisson family with robust varying, determining associations between sociodemographic, clinical, and booster dose perception variables.

**Results.** Data from 924 adults were analyzed. The IBV of the booster doses was 88.1%. A higher prevalence was associated with being male (aPR = 1.05; 95% CI [1.01–1.10]), having a good perception of efficacy and protective effect (PR = 3.69; 95% CI [2.57–5.30]) and belonging to the health sector (PR = 1.10; 95% CI [1.04–1.16]). There was greater acceptance of the recommendation of physicians and other health professionals (aPR = 1.40; 95% CI [1.27–1.55]).

**Conclusions.** Factors associated with higher IBV with booster dose include male gender, health sciences, physician recommendation, and good perception of efficacy.

## INTRODUCTION

The COVID-19 pandemic, which emerged in late December 2019 in Wuhan, China (*Poland, 2020*), continues to impact people around the world, with more than 755 million confirmed cases and more than 6.8 million deaths reported to the World Health Organization (WHO) as of February 10, 2023 (*World Health Organization, 2023a*). Despite the US Food and Drug Administration's approval of two types of treatments available to date, which are antiviral treatment with remdesivir and immunomodulatory therapy with barcitinib and tocilizumab, vaccination remains one of the best health protection measures to prevent COVID-19 (*Wang et al., 2020*; *U.S. Food and Drug Administration, 2022*). In the context of the current pandemic, vaccines were developed to lessen the impact of infection by the new coronavirus (*Urrunaga-Pastor et al., 2021*).

However, the success of vaccination programs depends on the efficacy and safety of the vaccine, and on the willingness of individuals to be vaccinated (*Sallam, 2021*; *Neumann-Böhme et al., 2022*). This disposition can be negatively affected by various sociodemographic and sociological elements, such as political trends, which poses a major challenge to global efforts to prevent and control infection (*Herrera-Añazco et al., 2021a*). The willingness of people to be vaccinated varies and includes acceptance, hesitancy or total refusal of the vaccine. In public health, it is very important to intervene with people who are hesitant because, unlike people with complete refusal, it is more likely that through good communication, understanding of their doubts and concerns, as well as taking into account the country context and the limitations of access to health services, it is possible to change to a state of acceptance of the vaccine (*Dubé & MacDonald, 2022*). This issue has led to studies around the world on the intention to be vaccinated (IBV) against COVID-19, among which a systematic review showed the existence of regional variability, indicating uptake rates below 60% in the Middle East, Eastern Europe and Russia (*Sallam, Al-Sanafi & Sallam, 2022*), and high uptake rates above 90% in East and Southeast Asia (*Sallam, 2021*), representing a problem in the successful implementation of vaccination programs globally.

One of the places most affected by the COVID-19 pandemic was Latin America and the Caribbean, finding a relatively high IBV with the first scheme (80%), with the highest prevalence in Mexico (88%) and the lowest in Haiti (43%) (*Urrunaga-Pastor et al., 2021*; *Sallam, 2021*). The opposite is true for the booster dose, since a study conducted in this population between February and March 2022 found that 33.7% of participants had not received the COVID-19 booster dose (*Urrunaga-Pastor et al., 2022*). Vaccination in Peru against COVID-19 began in February 2021, prioritizing the immunization of health personnel, then, in the following months, continued by age groups from oldest to youngest, namely "80 years and older", "70 to 79 years", "60 to 69 years" until reaching the age of "12 to 19 years", who were also considered, in the same year, as a target group for immunization (*Ministry of Health of Peru, 2023b*; *Sociedad Peruana de Urología, 2023*). In

Peru, IBV with the first dose was found to be 74.9% (*Sallam, Al-Sanafi & Sallam, 2022*). One of the primary reasons for not receiving the vaccine is the fear of potential adverse effects of immunization (*Urrunaga-Pastor et al., 2021*). Similarly, research on IBV with booster doses in the same country found that, as of April 2022, 21.5% had not receive a booster dose (*Bendezu-Quispe et al., 2022*). The Peruvian health situation room, as of February 05, 2023, has reported 4,482,582 confirmed cases of COVID-19 and 219,195 deaths (*Covid 19 en el Perú Ministerio de Salud, 2023*). According to the vaccination schedule of the Repositorio Único Nacional de Información en Salud (REUNIS), a total of 87,496,729 doses of vaccines have been applied until February 12, 2023, with coverage of the first and second doses greater than 90%, indicating a high acceptance of primary vaccination, probably because the ravages and consequences experienced by the pandemic in Peru were one of the most devastating in Latin America (*Garcia et al., 2020*). However, the third and fourth doses on the same date have a coverage of 74.1% and 26.0%, respectively, which probably implies a lower acceptance of the booster dose (*Ministry of Health of Peru, 2023a*). Although research related to IBV with both first and booster doses has been conducted in our country, we consider that the vaccination campaign needs continuous monitoring in order to identify factors that can be intervened to improve vaccination in Peru. Therefore, the objective of the present study was to explore the factors associated with the intention to be vaccinated with the booster dose against COVID-19 in Peru.

## MATERIALS & METHODS

### Study design
We conducted a cross-sectional study in Peru.

### Population and sample
The population comprised people over 18 years old who had received a minimum of 2 doses of the COVID-19 vaccine and who were living in Peru at the time of the survey. This was verified in the informed consent form with the following question: "Do you provide your informed consent to voluntarily participate in this research by declaring that you are 18 years of age or older and have at least 2 doses of the vaccine against COVID-19?". To ensure that participants resided in Peru, they were asked in the survey about the current department of residence. The sample was calculated with the STATA v.16.0 program (StataCorp LP, College Station, TX, USA), with a test of proportions for 2 samples, with a difference of 11% in the IBV between the population living in the city and the rural area according to background (*Herrera-Añazco et al., 2021a*). We used a power of 80%, a *p*-value of 0.05, and a design effect of 1.4. We estimated a 15% loss rate, estimating a minimum of 897 participants in total. Patient selection was by convenience sampling.

### Study procedure
A virtual survey form using Google Forms was developed in Spanish and distributed across different departments of Peru through the social networks Facebook and WhatsApp with the help of collaborators during the period from July 1 to 16, 2022. Likewise, physical surveys were conducted in public places without providing any incentives to the study

participants. The physical survey should be understood as the survey carried out by qualified personnel, authors and collaborators, and self-administered by the participant himself when he requested it. To control whether an individual accessed the questionnaire twice, each participant was asked to respond individually and only once, and socio-demographic aspects were also checked to search for duplicates. The virtual questionnaire had mandatory questions that did not allow continuing to the next questions. The physical surveys did not have mandatory questions to continue. The questionnaire was available for two weeks and no specific time was established for its completion. There was no follow-up, and no reminders were sent. At the end of this period, the databases of each collaborator were downloaded in Microsoft Excel 2016 format for their respective processing. It is important to mention that a previous pilot study was not carried out.

## Data collection instruments

Data was collected by means of an online survey consisting of 35 questions. The survey consisted of a main interface detailing the purpose of the study, including informed consent, and three sections: the first (16 items) assessed general characteristics and sociodemographic factors; the second (eight items) explored booster dose perception factors; and the third (11 items), focused onclinical factors. We adapted these questions from the survey conducted by the University of Maryland, USA, adding the word "booster" for the questions related to primary vaccination (available at: https://umdsurvey.umd.edu/jfe/form/ SV_6DnpVXXm2aYdnSe?token=VtALR8zAmXZUCli5Ud&Q_CreateFormSession=1& Q_Language=ES), which was conducted *via* Facebook (*Neumann-Böhme et al., 2022*).

In the last section, to measure depressive symptoms, the subscale of the DASS-21 instrument was used, in which each item responds according to the presence and intensity of each depressive symptom in the last week on a Likert-format response scale, ranging from 0 ("It did not happen to me") to 3 points ("It happened to me a lot or most of the time"). The corresponding subscale presents seven items in total, and its score is calculated with the sum of them. The DASS-21 is a reliable instrument with adequate construct validity and solid internal consistency (*Antúnez & Vinet, 2012*).

The survey questionnaire underwent a face validation process with expert judgment, including Likert-format scale options to assess coherence, clarity, and relevance. The V-Aiken calculation was used to estimate the questionnaire's main objective. If the calculation resulted in an estimate greater than 0.75 for coherence, clarity, and relevance, survey participants were then selected.

We conducted a *post-hoc* analysis to assess the internal structure using Chronbach's alpha of the items in the Stata statistical software with the alpha function.

## Dependent variable

The dependent variable was IBV, which was measured with the question "Do you agree to apply a booster dose to COVID-19? (Booster dose is understood as the third or fourth dose)". This question had four possible answers: "Definitely yes", "Probably yes", "Probably no", "Definitely no". The first two responses were grouped as positive IBV, while the last two as negative IBV. This categorization was made according to a

previous study (*Herrera-Añazco et al., 2021a*). We grouped the IBV of the first and second booster doses. In addition, those who had already received 2 booster doses were considered IBV positive.

## Independent variables

Some questions were categorized according to the following antecedents (*Herrera-Añazco et al., 2021a*), while others were categorized at the discretion of the researchers:

### Sociodemographic factors

We considered the variables sex, age, region, area of residence, religion, economic insecurity (defined as family income less than the minimum wage in Peru, *i.e.,* less than 1,025 soles ($271.27) (*El Peruano, 2023*)), have children, education (highest academic level completed), employment status, and food insecurity. This last variable was assessed with the following question: "Are you worried about having enough food for the next week?". The responses to this question were "Very worried", "Somewhat worried", "Not very worried", "Not at all worried". To obtain an accurate measure of food insecurity, we considered the first two responses as "Yes" and the last two as "No".

### Clinical factors

We evaluated depressive symptomatology using the depression subscale of the DASS-21. We considered with the presence of depressive symptoms those people who had a score higher than 5, this cut-off point has a sensitivity of 88.5% and a specificity of 86.8% (*Román, Santibáñez & Vinet, 2016*). We assessed the presence of comorbidities, defined as a story of cardiac involvement (such as myocarditis, pericarditis, deep vein thrombosis, myocardial infarction, or pulmonary embolism) and/or pulmonary involvement (such as respiratory failure, acute respiratory syndrome, pulmonary fibrosis, asthma, or pneumonia).

We evaluated the presence of events suspected to be attributed to vaccination or immunization (ESAVI) during the application of any of the doses against COVID-19 according to the participant's self-report. The variable was grouped into three categories: none, mild-moderate (such as general malaise, headache, pain at the injection site, among others) and severe (including people who reported having had an allergic reaction or requiring hospitalization). Additionally, we evaluated symptomatology suspicious for COVID-19. This was defined as experiencing three or more of the following symptoms 24 h prior to the survey: shortness of breath, cough, chest pain, fever, sore throat, eye pain, headache, fatigue, nausea, muscle pain or loss of smell. These criteria were described by WHO (*World Health Organization, 2023b*) and have already been used in other research (*Herrera-Añazco et al., 2021b*).

### Vaccine perception factors

We assessed the perception of the efficacy and protective effect of the booster dose with the following question: "Do you think the booster dose is effective and confers a protective effect against COVID-19?". This question had four possible answers: "Definitely yes", "Probably yes", "Probably no", "Definitely no". The first two responses were grouped as "Yes", while the last two as "No" (lack of confidence about the efficacy of the booster dose).

We evaluated concerns of experiencing adverse effects from the booster dose with the following question: "How concerned do you feel about experiencing adverse effects from the booster dose? (Booster dose is understood as the third or fourth dose)". This question had four possible responses: "Very concerned, "Moderately concerned", "Not at all concerned", "Not at all concerned". The first two answers were grouped as "Yes", while the last two as "No".

We assesed the IBV based on the recommendations of various agents, including friends and relatives, physicians and other health professionals, government health authorities such as Ministerio de Salud (MINSA), and politicians. For this purpose, we elaborated a generic question: "How likely are you to accept the booster dose against COVID-19 if it were recommended to you...?". Complementing with the various agents mentioned above. The possible answers were three: "More likely", "About the same" and "Less likely". The first answer indicated a good acceptance of the recommendation of that agent, while the last two were grouped together, indicating a lower acceptance or lack of influence of that agent on the recommendation to be vaccinated with the booster dose.

## Statistical analysis

Statistical analysis was performed in Stata 16.0 (StataCorp, College Station, TX, USA). We categorized all variables and described absolute and relative frequencies. For the bivariate analysis, we used the chi-square test to evaluate the associations of each categorical variable with the IBV after evaluating the assumption of expected frequency. For both the bivariate and multiple regression model, we used the Poisson family generalized linear model function with robust variance and log-link function. Prevalence ratios with 95% confidence intervals were estimated, and significance was set at $p < 0.05$ for two tails.

The multiple regression analysis was constructed using the statistical model with the stepwise test with forward selection and a criterion of $p < 0.20$, including the variables sex, age, depressive symptoms, comorbidities, acceptance of the booster dose on the recommendation of friends and family, physicians and other health professionals, MINSA, fear of adverse events and perception of efficacy. This last variable was removed from the final model because it presented collinearity with the acceptance of the booster dose on the recommendation of physicians and other health professionals. Collinearity was assessed using the variance inflation factor.

## Ethical aspects

The research protocol was reviewed and approved by the Institutional Research Ethics Committee of the Avendaño Clinic, Lima, Peru, with the code (031-2022-CIEI). The ethical principles of the Declaration of Helsinki of beneficence, autonomy, and non-maleficence were respected. The questionnaire was anonymous and included a declaration of virtual informed consent for the online surveys. For physical surveys, participants provided their written informed consent through their signature. In both cases, we specified the objective of the study, the use of their data and assured them of anonymity throughout the research. As observational methods were used, our study did not generate greater risk for the participants.

## RESULTS

### Characteristics of the study sample

All items received a favorable expert judgment evaluation (V-Aiken > 0.75), and an acceptable internal confidence level was obtained with an alpha of 0.86.

A total of 934 persons completed the survey. We excluded 10 respondents, including two duplicates, 5 minors and 3 due to informed consent (refusal to participate or having less than two doses). This resulted in an analyzed sample of 924 people, with 91.7% responding through virtual questionnaires and 8.3% through physical surveys. No significant differences were observed between the groups concerning the intention to be vaccinated ($p = 0.759$), so both subpopulations were merged into one. Of the participants, 53.7% ($n = 496$) were female, 66.5% ($n = 614$) were aged between 18 and 29 years, 87.1% ($n = 805$) lived in an urban area and 62.0% ($n = 573$) lived in the Coastal region. Additionally, 42.8% ($n = 395$) reported being food insecure, while 29.9% ($n = 276$) had economic insecurity; participants who reported having children was 31.8% ($n = 294$) (Table 1A). In addition, 26.0% ($n = 240$) presented depressive symptoms and 23.2% ($n = 214$) reported having comorbidities. The probability of acceptance of vaccination with the booster dose was 66.6% ($n = 615$) on the recommendation of physicians, 54.0% ($n = 499$) on the recommendation of government health authorities (MINSA), 43.6% ($n = 384$) on the recommendation of family and friends, and 17.2% ($n = 159$) on the recommendation of politicians. In addition, 71.7% ($n = 662$) reported some mild-moderate ESAVI, 10.8% reported symptomatology suspicious for COVID-19 and 86.3% ($n = 797$) of the participants had three doses of vaccine at the time of the survey. Likewise, 88.1% ($n = 814$) affirmed IBV (Table 1B). The IBV with the booster dose varied by department, with the following percentages: Junín (93.8%), Amazonas (91.8%), Lima (90.5%), Lambayeque (87.9%), San Martín (87.9%), Cajamarca (85.0%) and Piura (83.2%).

While, the data from non-responders cannot be quantified, an interim analysis of the physical survey non-responders indicated a percentage of less than 10%. The comparison between the population that participated in the physical and virtual surveys did not differ (see Table S1).

### Bivariate analysis according to intention to be vaccinated (IBV) with the booster dose

Several variables were found to be significantly associated with the intention to be vaccinated (IBV), as detailed in Table 2. These variables include age ($p = 0.002$), having children ($p = 0.002$), being related to the health sector ($p = 0.001$), perception of the efficacy and protective effect of the booster dose (<0.001), as well as fear of adverse effects ($p < 0.001$) were significantly associated with the IBV (Table 2). Additionally, bivariate analysis revealed a significantly association ($p < 0.001$) between IBV and the recommendation of physicians, government health authorities (MINSA), family and friends, and politicians (Table 2).

**Table 1** (A) Sociodemographic characteristics of the booster dose COVID-19 in Peru, July 2022 (*n* = 924).

| Characteristics | N (%) |
|---|---|
| Sex | |
|     Female | 496 (53.7) |
|     Male | 426 (46.1) |
|     Missing | 2 (0.2) |
| Age (years) | |
|     18–29 | 614 (66.5) |
|     30–44 | 184 (19.9) |
|     45 years or older | 125 (13.5) |
|     Missing | 1 (0.1) |
| Area of residence | |
|     Urban | 805 (87.1) |
|     Rural | 117 (12.7) |
|     Missing | 2 (0.2) |
| Region | |
|     Coast | 573 (62.0) |
|     Highlands | 201 (21.8) |
|     Jungle | 150 (16.2) |
| Religion | |
|     Catholic | 701 (75.9) |
|     Evangelical | 121 (13.1) |
|     Atheist or agnostic | 47 (5.1) |
|     Other religion | 54 (5.8) |
|     Missing | 1 (0.1) |
| Food insecurity | |
|     No | 525 (56.8) |
|     Yes | 395 (42.8) |
|     Missing | 4 (0.4) |
| Economic insecurity | |
|     No | 647 (70.0) |
|     Yes | 276 (29.9) |
|     Missing | 1 (0.1) |
| Have children | |
|     No | 629 (68.1) |
|     Yes | 294 (31.8) |
|     Missing | 1 (0.1) |
| Education | |
|     Primary school completed or less | 22 (2.4) |
|     High school completed | 360 (39.0) |
|     College, pre-college, or university | 475 (51.4) |
|     University postgraduate degree complete | 66 (7.1) |
|     Missing | 1 (0.1) |

**Table 1** (*continued*)

| Characteristics | N (%) |
|---|---|
| Employment status | |
| Not related to health sciences | 401 (43.4) |
| In relation to health sciences | 466 (50.4) |
| Not working, not studying | 54 (5.9) |
| Missing | 3 (0.3) |

**(B) Clinical and perception characteristics of the booster dose COVID-19 in Peru, July 2022 ($n = 924$).**

| | |
|---|---|
| Depressive symptoms | |
| No | 681 (73.7) |
| Yes | 240 (26.0) |
| Missing | 3 (0.3) |
| Comorbidities | |
| No | 710 (76.8) |
| Yes | 214 (23.2) |
| ESAVI | |
| None | 249 (26.9) |
| Mild-moderate | 662 (71.7) |
| Severe | 13 (1.4) |
| Symptomatology suspicious of COVID-19 | |
| No | 823 (89.1) |
| Yes | 100 (10.8) |
| Missing | 1 (0.1) |
| Likelihood of acceptance of booster dose on the recommendation of family and friends | |
| Negative influence/Indifferent | 490 (56.0) |
| Positive influence | 384 (43.6) |
| Missing | 50 (5.4) |
| Likelihood of acceptance of booster dose at the recommendation of physicians and other health professionals | |
| Negative influence/Indifferent | 259 (28.0) |
| Positive influence | 615 (66.6) |
| Missing | 50 (5.4) |
| Probability of acceptance of the booster dose upon the recommendation of government health authorities (MINSA) | |
| Negative influence/Indifferent | 374 (40.5) |
| Positive influence | 499 (54.0) |
| Missing | 51 (5.5) |
| Likelihood of acceptance of booster dose at the recommendation of politicians | |
| Negative influence/Indifferent | 714 (77.3) |
| Positive influence | 159 (17.2) |
| Missing | 51 (5.5) |
| Perception of the efficacy and protective effect of the booster dose | |
| No | 86 (9.3) |
| Yes | 838 (90.7) |

**Table 1** (*continued*)

**(B) Clinical and perception characteristics of the booster dose COVID-19 in Peru, July 2022 ($n = 924$).**

| | |
|---|---|
| Fear of adverse effects of the booster dose | |
| No | 479 (51.8) |
| Yes | 445 (48.2) |
| Vaccine doses | |
| 2 doses | 69 (7.5) |
| 3 doses | 797 (86.3) |
| 4 doses | 50 (5.4) |
| Missing | 8 (0.8) |
| Intention to vaccinate with the booster dose | |
| No | 110 (11.9) |
| Yes | 814 (88.1) |

**Notes.**
[a] Some variables may sum to less than 924 due to missing data.

## Factors associated with intention to be vaccinated (IBV) with the booster dose

In the simple regression, being related to health (PR = 1.10; 95% CI [1.04–1.16]; $p < 0.001$), having a good perception of its efficacy and protective effect (PR = 3.69; 95% CI [2.57–5.30]; $p < 0.001$), the recommendation of family and friends (PR = 1.28; 95% CI [1.22–1.34]; $p < 0.001$), physicians (PR = 1.52; 95% CI [1.38–1.66]; $p < 0.001$), government health authorities (MINSA) (PR = 1.32; 95% CI [1.24–1.40]; $p < 0.001$) and politicians (PR = 1.17; 95% CI [1.13–1.21]; $p < 0.001$) were associated with a higher IBV. Conversely, being aged 45 or older (PR = 0.89; 95% CI [0.81–0.97]; $p = 0.011$), having children (PR = 0.92; 95% CI [0.87–0.98]; $p = 0.006$) and being afraid of adverse effects (PR = 0.91; 95% CI [0.87–0.96]; $p < 0.001$), were associated with a lower prevalence of IBV with the booster dose. In the adjusted regression, being male (aPR = 1.05; 95% CI [1.01–1.10]; $p = 0.024$), recommendation by family and friends (aPR = 1.08; 95% CI [1.04–1.12]; $p < 0.001$), and recommendation by physicians (aPR = 1.40; 95% CI [1.27–1.55]; $p < 0.001$) were associated with a higher prevalence of IBV with the booster dose, while age 45 years and older was associated with lower IBV (aPR = 0.89; 95% CI [0.81–0.98]; $p = 0.018$) (Table 3).

## DISCUSSION

In our study, a high IBV with the booster dose has been observed (88.1%). This figure aligns with similar findings in other parts of the world, where participants have a strong willingness to receive the additional dose with values of 87% of adults in Denmark (*Jørgensen, Nielsen & Petersen, 2022*), 82.5% of adults in Iran (*Askarian et al., 2023*), 84.6% of adults in Bangladesh (*Roy, Azam & Islam, 2023*), 81.2% individuals with chronic diseases in Malaysia (*Abdullah, Ching & Ali, 2023*) and 82.8% in elderly individuals in China (*Qin et al., 2022b*). Furthermore, our results are comparable to those reported in Chile (88.2%). However, this result contrasts significantly with the situation in Ghana, where only 46.2% of adults have a high intention to be vaccinated (IBV) (*Storph et al., 2023*). However, a systematic review indicated that the IBV of the booster dose is different from the actual

**Table 2** Bivariate analysis of study characteristics according to intention to be vaccinated with the booster dose COVID-19 in Peru, July 2022 ($n = 924$).

| Characteristics | Intention to be vaccinated | | *p* |
|---|---|---|---|
| | No ($n = 110$) $n$(%) | Yes ($n = 814$) $n$(%) | |
| Sex[a] | | | 0.436 |
|   Female | 63 (12.7) | 433 (87.3) | |
|   Male | 47 (11.0) | 379 (89.0) | |
| Age (years)[a] | | | 0.002 |
|   18–29 | 57 (9.3) | 557 (90.7) | |
|   30–44 | 29 (15.8) | 155 (84.2) | |
|   45 years or older | 24 (19.2) | 101 (80.8) | |
| Area of residence[a] | | | 0.751 |
|   Urban | 95 (11.8) | 710 (88.2) | |
|   Rural | 15 (12.8) | 102 (87.2) | |
| Religion[a] | | | 0.152 |
|   Catholic | 74 (10.6) | 627 (89.4) | |
|   Evangelical | 20 (16.5) | 101 (83.5) | |
|   Atheist or agnostic | 8 (17.0) | 39 (83.0) | |
|   Other religion | 8 (14.8) | 46 (85.2) | |
| Food insecurity[a] | | | 0.869 |
|   No | 63 (12.0) | 462 (88.0) | |
|   Yes | 46 (11.7) | 349 (88.3) | |
| Economic insecurity[a] | | | 0.674 |
|   No | 79 (12.2) | 568 (87.8) | |
|   Yes | 31 (11.2) | 245 (88.8) | |
| Have children[a] | | | 0.002 |
|   No | 61 (9.7) | 568 (90.3) | |
|   Yes | 49 (16.7) | 245 (83.3) | |
| Education[a] | | | 0.356 |
|   Primary school completed or less | 4 (18.2) | 18 (81.8) | |
|   High school completed | 35 (9.7) | 325 (90.3) | |
|   College, pre-college, or university | 62 (13.1) | 413 (86.9) | |
|   University postgraduate degree complete | 9 (13.6) | 57 (86.4) | |
| Employment status[a] | | | 0.001 |
|   Not related to health sciences | 65 (16.3) | 335 (83.7) | |
|   In relation to health sciences | 37 (7.9) | 429 (92.1) | |
|   Not working, not studying | 8 (14.8) | 46 (85.2) | |
| Depressive symptoms[a] | | | 0.938 |
|   No | 81 (11.9) | 600 (88.1) | |
|   Yes | 29 (12.1) | 211 (87.9) | |
| Comorbidities | | | 0.183 |
|   No | 79 (11.1) | 631 (88.9) | |
|   Yes | 31 (14.5) | 183 (85.5) | |

| Characteristics | Intention to be vaccinated | | *p* |
|---|---|---|---|
| | **No (*n* = 110)** *n*(%) | **Yes (*n* = 814)** *n*(%) | |
| ESAVI | | | 0.191 |
| None | 35 (14.1) | 214 (85.9) | |
| Mild-moderate | 72 (10.9) | 590 (89.1) | |
| Severe | 3 (23.1) | 10 (76.9) | |
| Symptomatology suspicious of COVID-19 | | | 0.764 |
| No | 99 (12.0) | 724 (88.0) | |
| Yes | 11 (11.0) | 89 (89.0) | |
| Likelihood of acceptance of booster dose on the recommendation of family and friends[a] | | | <0.001 |
| Negative influence/Indifferent | 108 (22.0) | 382 (78.0) | |
| Positive influence | 2 (0.5) | 382 (99.5) | |
| Likelihood of acceptance of booster dose at the recommendation of physicians and other health professionals[a] | | | <0.001 |
| Negative influence/Indifferent | 93 (35.9) | 166 (64.1) | |
| Positive influence | 17 (2.8) | 598 (97.2) | |
| Probability of acceptance of the booster dose upon the recommendation of government health authorities (MINSA)[a] | | | <0.001 |
| Negative influence/Indifferent | 97 (25.9) | 277 (74.1) | |
| Positive influence | 13 (2.6) | 486 (97.4) | |
| Likelihood of acceptance of booster dose at the recommendation of politicians[a] | | | <0.001 |
| Negative influence/Indifferent | 109 (15.3) | 605 (84.7) | |
| Positive influence | 1 (0.6) | 158 (99.4) | |
| Perception of the efficacy and protective effect of the booster dose | | | <0.001 |
| No | 64 (74.4) | 22 (25.6) | |
| Yes | 46 (5.5) | 792 (94.5) | |
| Fear of adverse effects of the booster dose | | | <0.001 |
| No | 38 (7.9) | 441 (92.1) | |
| Yes | 72 (16.2) | 373 (83.8) | |

**Notes.**
[a] Some variables may add up to less than 924 due to missing data. Chi-square test was used for all variables.

vaccine uptake (*Abdelmoneim et al., 2022*) . Furthermore, rates of acceptance and hesitation can exhibit variability depending on contextual factors and the dynamics of epidemic waves (*Noh et al., 2022*). The high IBV in our study probably has two reasons. First, the positive IBV included those who responded in the survey as ''definitely yes'' and ''probably yes'' to be vaccinated, indicating a range of willingness rather than absolute acceptance of the booster dose. Secondly, the high IBV may be attributed to the timing of our data collection, occurring one week after the onset of the fourth wave of COVID-19 in Peru (*Salud Con Lupa, 2022*).

**Table 3   Factors associated with the intention to be vaccinated with the COVID-19 booster dose in Peru, July 2022.**

| Characteristics | Bivariate analysis[*] | | | Multiple regression[**] | | |
|---|---|---|---|---|---|---|
| | cPR | 95% CI | p | aPR | 95% CI | p |
| Sex | | | | | | |
| Female | Ref. | | | Ref. | | |
| Male | 1.02 | 0.97–1.07 | 0.434 | 1.05 | 1.01–1.10 | 0.024 |
| Age (years) | | | | | | |
| 18–29 | Ref. | | | Ref. | | |
| 30–44 | 0.93 | 0.87–0.99 | 0.031 | 0.94 | 0.88–1.01 | 0.076 |
| 45 years or older | 0.89 | 0.81–0.97 | 0.011 | 0.89 | 0.81–0.98 | 0.018 |
| Area of residence | | | | | | |
| Urban | Ref. | | | | | |
| Rural | 0.99 | 0.92–1.06 | 0.758 | | NA | |
| Food insecurity | | | | | | |
| No | Ref. | | | | | |
| Yes | 1.00 | 0.96–1.05 | 0.869 | | NA | |
| Economic insecurity | | | | | | |
| No | Ref. | | | | | |
| Yes | 1.01 | 0.96–1.06 | 0.670 | | NA | |
| Have children | | | | | | |
| No | Ref. | | | | | |
| Yes | 0.92 | 0.87–0.98 | 0.006 | | NA | |
| Education | | | | | | |
| College, pre-college, or university | Ref. | | | | | |
| Primary school completed or less | 0.94 | 0.77–1.15 | 0.552 | | NA | |
| High school completed | 1.04 | 0.99–1.09 | 0.130 | | | |
| University postgraduate degree complete | 0.99 | 0.90–1.10 | 0.897 | | | |
| Employment status | | | | | | |
| Not related to health sciences | Ref. | | | | | |
| In relation to health sciences | 1.10 | 1.04–1.16 | <0.001 | | NA | |
| Not working, not studying | 1.02 | 0.90–1.15 | 0.780 | | | |
| Depressive symptoms | | | | | | |
| No | Ref. | | | Ref. | | |
| Yes | 1.00 | 0.94–1.05 | 0.938 | 1.03 | 0.97–1.08 | 0.315 |
| Comorbidities | | | | | | |
| No | Ref. | | | Ref. | | |
| Yes | 0.96 | 0.91–1.02 | 0.216 | 0.97 | 0.91–1.02 | 0.257 |
| ESAVI | | | | | | |
| None | Ref. | | | Ref. | | |
| Mild-moderate | 1.04 | 0.98–1.10 | 0.211 | 1.06 | 1.00–1-12 | 0.035 |
| Severe | 0.90 | 0.66–1.21 | 0.472 | 0.88 | 0.62–1.24 | 0.461 |

**Table 3** (*continued*)

| Characteristics | Bivariate analysis[*] | | | Multiple regression[**] | | |
|---|---|---|---|---|---|---|
| | cPR | 95% CI | *p* | aPR | 95% CI | *p* |
| Symptomatology suspicious of COVID-19 | | | | | | |
| No | Ref. | | | Ref. | | |
| Yes | 1.01 | 0.94–1.09 | 0.756 | 0.98 | 0.91–1.05 | 0.528 |
| Likelihood of acceptance of booster dose on the recommendation of family and friends | | | | | | |
| Negative influence/Indifferent | Ref. | | | Ref. | | |
| Positive influence | 1.28 | 1.22–1.34 | <0.001 | 1.08 | 1.04–1.12 | <0.001 |
| Likelihood of acceptance of booster dose at the recommendation of physicians and other health professionals | | | | | | |
| Negative influence/Indifferent | Ref. | | | Ref. | | |
| Positive influence | 1.52 | 1.38–1.66 | <0.001 | 1.40 | 1.27–1.55 | <0.001 |
| Probability of acceptance of the booster dose upon the recommendation of government health authorities (MINSA) | | | | | | |
| Negative influence/Indifferent | Ref. | | | Ref. | | |
| Positive influence | 1.32 | 1.24–1.40 | <0.001 | 1.05 | 0.99–1.11 | 0.124 |
| Likelihood of acceptance of booster dose at the recommendation of politicians | | | | | | |
| Negative influence/Indifferent | Ref. | | | | | |
| Positive influence | 1.17 | 1.13–1.21 | <0.001 | | NA | |
| Perception of the efficacy and protective effect of the booster dose | | | | | | |
| No | Ref. | | | | | |
| Yes | 3.69 | 2.57–5.30 | <0.001 | | NA[a] | |
| Fear of adverse effects of the booster dose | | | | | | |
| No | Ref. | | | Ref. | | |
| Yes | 0.91 | 0.87–0.96 | <0.001 | 0.96 | 0.92–1.00 | 0.060 |

**Notes.**

95% CI, 95% confidence interval; cPR, crude prevalence ratio; aPR, adjusted prevalence ratio; NA, not supported in the multiple regression model.

[*]Generalized linear models of the Poisson family with robust variances and log link were used.

[**]Generalized linear models of the Poisson family with robust variances and log link with stepwise test and forward selection (*p* < 0.20) were used.

[a]Not supported because of collinearity with the probability of acceptance of the booster dose on the recommendation of physicians and other health professionals.

Our results indicate that having a good perception of the effectiveness of the booster dose is strongly associated with a higher IBV (PR:3.69). Studies conducted in China (*Qin et al., 2022b*; *Tung et al., 2019*; *Hu et al., 2022*), Malaysia (*Abdullah, Ching & Ali, 2023*) and Czech Republic (*Klugar et al., 2021*), as well as a systematic review (*Abdelmoneim et al., 2022*), indicated that the perceived efficacy or effectiveness of vaccines was associated with IBV of the booster dose. On the other hand, studies conducted in China (*Lai et al., 2021*; *Qin et al., 2022*) and countries such as Poland (*Rzymski, Poniedziałek & Fal, 2021*), Italy (*Folcarelli et al., 2022*), Greece (*Galanis et al., 2022*), and Algeria (*Lounis et al., 2022*) reported that the reasons for refusing to receive the booster dose were uncertainty about its efficacy and the perception that an additional dose was not necessary. It should also be considered that these studies were conducted at different times during the COVID-19

pandemic, which could result in variations in the results, as the perception of the population may have changed over time.

On the other hand, the crude regression analysis indicated that fear of adverse effects of the vaccine is one of the reasons for a lower IBV. This aligns with many studies that concluded that concern about adverse effects was the main factor for refusal or hesitancy to receive a booster dose (*Qin et al., 2022*; *Rzymski, Poniedziałek & Fal, 2021*; *Folcarelli et al., 2022*; *Galanis et al., 2022*; *Lounis et al., 2022*). However, in the adjusted model, no such association was observed, possibly because this effect may have been overshadowed by the recommendation of physicians and other health professionals. It seems that these recommendations may carry more weight than the fear of adverse effects of the vaccine. Despite various studies supporting the efficacy and safety of the booster dose in preventing COVID-19 (*Bar-On et al., 2021*; *Bar-On et al., 2021*; *Barda et al., 2021*; *Moreira et al., 2022*) and its variants (*Chenchula et al., 2022*), achieving good communication and literacy in the population is crucial. This is particularly important when considering the situational context of each country and locality, addressing their main doubts and concerns about the booster vaccine (*Dubé & MacDonald, 2022*; *Lazarus et al., 2021*). Such efforts are vital for improving vaccination rates in many parts of the world. Our study showed that being a man, compared to being a woman, was associated with a higher IBV. These results are similar with findings in Italy (*Miragliadel Giudice et al., 2023*), Iran (*Askarian et al., 2023*) and in a meta-analysis (*Abdelmoneim et al., 2022*). This could be explained by the higher morbidity and mortality against COVID-19 that men had in Peru (*Ramírez-Soto, Ortega-Cáceres & Arroyo-Hernández, 2022*), also because women are more susceptible to vaccine conspiracy beliefs (*Sallam et al., 2021*). However, other studies have shown even higher IBV in women, probably because they are more tolerable to "pandemic fatigue" (*Urrunaga-Pastor et al., 2022*; *Bendezu-Quispe et al., 2022*; *Rzymski, Poniedziałek & Fal, 2021*; *Miao et al., 2022*).

Regarding age, the only category that was associated with IBV was age 45 years and older, obtaining a lower prevalence compared to age 18 to 29 years. This aligns with findings from other studies that indicate a higher IBV with booster dose as evidenced in Australia (*Allen et al., 2022*) and Algeria (*Lounis et al., 2022*). Perhaps, these results can be attributed to a preference bias, as older adults saw early efficacy to early vaccine exposure, because they were prioritized with primary vaccination for having been the most affected at the beginning of the pandemic (*Valladares-Garrido et al., 2022*). Additionally, another study notes that middle-aged and older age groups may have greater concern for adverse events with an additional dose and thus a lower IBV (*Qin et al., 2022*). However, other research has reported that older adults have the highest IBV with the booster dose because they are a risk group (*Rzymski, Poniedziałek & Fal, 2021*; *Folcarelli et al., 2022*).

Additionally, in the multiple regression, we unexpectedly found a slightly higher probability of IBV was found in those who reported some mild-moderate ESAVI compared to those who did not report any. Although this may be due to the fact that people with mild-moderate ESAVI that is not life-threatening and does not cause major harm (such as fever, malaise, headache, muscle aches, *etc.*), have a better perception of vaccine efficacy because of the same side effects they suffer. However, our measurements may not be entirely correct, as the presence of ESAVI must be rigorously evaluated by a physician

or qualified health personnel (*Anonymous , 2023a*). In contrast to a previous study that evaluated primary vaccination (*Herrera-Añazco et al., 2021b*), we found no association between suspected COVID-19 symptomatology and increased IBV with the booster dose, however, it is possible that our results are consistent, since, despite being at the beginning of the fourth wave (*Salud Con Lupa, 2022*), we only observed an increase in the number of cases, but no hospitalizations or deaths, which may have generated a sense of security and may have overlooked the suspected COVID-19 symptomatology (*Ascarza, 2022*).

The present study indicated a higher probability of being vaccinated with the booster dose in people related to health sciences same results as Iran (*NeJhaddadgar et al., 2023*), possibly explained by a better handling of information about vaccines (*Bartoš et al., 2022*; *Paterson et al., 2016*), or by the higher risk of getting sick from COVID-19 (*Ing et al., 2020*), however, studies in this population are variable; there are some that mention that health personnel are willing to be vaccinated with the booster dose, while other studies show that there is a group of people related to the health area who are reluctant or unwilling to accept a booster dose (*Lounis et al., 2022*; *Dzieciolowska et al., 2021*; *Verdú-Victoria et al., 2022*). Our results are important in vaccination since it has been shown that a higher IBV with the booster dose in health personnel has been associated with a higher recommendation to their patients (*Paterson et al., 2016*; *Della Polla et al., 2022*).

Third, it is important to note that the results of the survey show the sources of information to know about the COVID-19 booster dose. Our study also reports a high likelihood of IBV with the booster dose on the recommendation of health care professionals such as physicians, a result that is consistent with other research that evaluated the acceptability of COVID-19 vaccination on the recommendation of physicians and health care providers (*Herrera-Añazco et al., 2021a*; *Folcarelli et al., 2022*; *Goldman et al., 2020*; *Reiter, Pennell & Katz, 2020*). A systematic review mentions that the influence of physicians are the most important factors for the population to accept vaccination with the booster dose, because health personnel are best placed to disseminate standardized, evidence-based messages (*Lin, Tu & Beitsch, 2021*). Additionally, a study conducted in China reported that 80.6% of its respondents reported that their physicians had an influence on them in making the decision to accept the vaccine (*Wang et al., 2020*). Likewise, the Ipsos report that evaluated the 2022 trust ranking in urban population with internet access between May and June, indicated that doctors and scientists are the most trusted professionals for Peruvians (*Ipsos Perú, 2022*). Therefore, it is very likely that the promotion of vaccination and IBV with the booster dose against COVID-19 falls mainly on health professionals. We also analyzed the recommendation of governmental health authorities such as MINSA, which is the entity in charge of directing health policies and actions in Peru (*Government of Peru, 2023*), finding a higher probability of IBV with the recommendation of this agency; however, in the adjusted analysis it was not statistically significant. These findings are particularly important because government health authorities are key players in achieving optimal vaccination and adequate literacy (*Lazarus et al., 2021*; *Biasio, 2017*). Several studies, including a survey that evaluated 19 countries in various regions of the world, indicate that people are more likely to be vaccinated if they trust their government authorities (*Folcarelli et al., 2022*; *Lazarus et al., 2021*; *Siegrist & Zingg, 2014*; *Lazarus et al., 2020*). In this way, we can notice

the great impact generated by physicians, who can influence the decisions of the population as well as their work as an indispensable entity in the prevention and promotion of health. Therefore, we recommend paying close attention to these results to use their influence on the diffusion of the vaccination not only against COVID-19, but even on other vaccines such as yellow fever, hepatitis B, or influenza (*Government of Peru, 2022*).

The results of our study indicate a greater acceptance of IBV in the face of the recommendation of politicians. This finding is significant, as *Herrera-Añazco et al. (2021a)* said, the media usually give a lot of advertising space to politicians, which could be used to promote vaccination. In addition, a high influence of politicians on IBV has been reported in several studies (*Urrunaga-Pastor et al., 2021*; *Lazarus et al., 2021*), however, this influence can be altered by several factors such as infodemic, doubts about vaccination, political instability, as well as the messages that politicians themselves post on their social networks (*Garcia et al., 2020*; *Borgesdo Nascimento et al., 2022*; *White et al., 2023*; *Malik et al., 2020*). Therefore, it is suggested that politicians transmit information based on scientific evidence and promote vaccination campaigns adapted to the different realities of health access in each country.

No differences were found in IBV in people of rural and urban residence. It is possible that the difference between the percentage of respondents living in urban areas (87.3%) with respect to those living in rural areas (12.7%) did not allow a true comparison. However, other studies have reported that living in a rural area is associated with lower IBV, such as in Guatemala and marginalized communities in the United Kingdom and the United States, with access to vaccination sites and language being factors that negatively influenced vaccination progress and thus booster dose (*Taylor, 2022*; *Halvorsrud et al., 2022*; *Dong et al., 2022*). Given that Peru is a country with a marked geographic variation, with many of its rural communities challenging access to health care (*Gozzer Infante, 2020*) and limited access to information and communication technologies (*Calderón, 2022*), a constant evaluation of vaccination in these sectors is recommended, especially in the departments of Huancavelica, Huánuco, Cajamarca, San Martín, Amazonas, Puno, Loreto, Madre de Dios, Ayacucho, Cusco, San Martín and Ucayali. As of February 12, 2023 they have the lowest vaccination coverage with the booster dose (*Ministry of Health of Peru, 2023a*).

Our research has certain limitations and there may be some biases, on the one hand, it is possible that there was a social desirability bias in the physical surveys, however, the staff tried to minimize this bias, ensuring their anonymity in each survey, having a friendly treatment with the person, asking in a neutral way when required or a little more indirectly when evaluating, for example, depressive symptoms. On the other hand, self-report bias, common in questionnaires of this type, could occur in virtual surveys; however, questions from validated questionnaires previously used in virtual surveys were used. As mentioned above, there were no statistically significant differences in the IBV between the physical and virtual surveys. Since we used non-random convenience sampling, it is likely that there was a selection bias that makes the results not extrapolated to the entire Peruvian population, so the results of the study should be taken with caution. In addition, there is a risk of representativeness in rural areas, since they represent only 12.8% of the respondents, as

well as in age, since 86.5% of our sample was under 45 years of age. Although the number of people who entered the final analysis is known, it was not possible to quantify the response bias because it was an open invitation and we did not know how many people in total received the survey. On the other hand, the depressive symptoms subscale of the DASS-21 instrument has a sensitivity of 88.5% (*Román, Santibáñez & Vinet, 2016*) and provides only a snapshot of recent mental health symptomatology, so depressive symptoms scores may not fully correlate with the clinical diagnosis made by a mental health professional. Additionally, the variables comorbidities and ESAVI were not adequately measured, as they were self-reported, likewise, COVID-19 suspicious symptomatology does not indicate disease, but symptoms compatible with the disease that were reported by the participant. Despite the limitations, this study establishes the association between sociodemographic variables and those related to the perception of the COVID-19 booster dose, which may contribute to the development of future strategies and improved immunization programs.

## CONCLUSIONS

Our study indicates a high IBV with the booster dose (88.1%), and there are factors associated with a higher IBV, including male gender, people related to health sciences, adults with a good perception of the effectiveness and protective effect of the booster dose, with the latter emerging as the most influential determinant. Likewise, our findings indicate that individuals aged 45 and older exhibit a lower IBV. Among all the agents studied, the recommendation of doctors emerged as the most influential on IBV while the recommendation of politicians exerted the least influence.

We recommend an increased dissemination of the safety and effectiveness of the vaccine on social networks, television and radio. Additionally, we suggest greater communication by doctors and health personnel towards patients who have doubts or fears about the booster vaccination. We recommend campaigns against fake news since these are usually shared among friends and family, influencing the IBV. Finally, efforts should be directed towards reaching at-risk populations, particularly older adults and people in remote areas without internet access.

**Abbreviations**

| | |
|---|---|
| **WHO** | World Health Organization |
| **IBV** | Intention to be vaccinated |
| **REUNIS** | Repositorio Único Nacional de Información en Salud |
| **ESAVI** | Events Suspected to be Attributed to Vaccination or Immunization |
| **MINSA** | Ministerio de Salud |
| **CI** | Confidence interval |
| **PR** | Prevalence ratio |
| **PRa** | Adjusted prevalence ratio |

## ACKNOWLEDGEMENTS

We would like to thank Milagros Lizbeth Garcia Cordova, Joel Emmanuel Inga Chero, Marisol Álvarez Ríos, Josias Caleb Flores Loly and Alicia Torres Mera, who participated in

the dissemination of the virtual form and data collection, for their kind collaboration. We would also like to thank all those who volunteered to participate in this study.

### Funding
The authors received no funding for this work.

### Competing Interests
The authors declare there are no competing interests.

### Author Contributions
- Rodrigo Camacho-Neciosup conceived and designed the experiments, performed the experiments, prepared figures and/or tables, authored or reviewed drafts of the article, and approved the final draft.
- Ericka N. Balcazar-Huaman conceived and designed the experiments, performed the experiments, prepared figures and/or tables, authored or reviewed drafts of the article, and approved the final draft.
- Margarita L. Alvarez-Vilchez performed the experiments, prepared figures and/or tables, authored or reviewed drafts of the article, and approved the final draft.
- Janith P. De la Cruz-Galán performed the experiments, authored or reviewed drafts of the article, and approved the final draft.
- Yubely Gálvez-Guadalupe performed the experiments, authored or reviewed drafts of the article, and approved the final draft.
- Edwin D. Garcia-Muñoz performed the experiments, authored or reviewed drafts of the article, and approved the final draft.
- Greysi Cerron-Daga performed the experiments, authored or reviewed drafts of the article, and approved the final draft.
- Virgilio E. Failoc-Rojas conceived and designed the experiments, performed the experiments, analyzed the data, prepared figures and/or tables, authored or reviewed drafts of the article, and approved the final draft.
- Mario J. Valladares-Garrido performed the experiments, analyzed the data, authored or reviewed drafts of the article, and approved the final draft.

### Human Ethics
The following information was supplied relating to ethical approvals (i.e., approving body and any reference numbers):

The research protocol was reviewed and approved by the Institutional Research Ethics Committee of the Avendaño Clinic, Lima, Peru

### Data Availability
The raw data are available in the Supplementary File.

## Supplemental Information

Supplemental information for this article can be found online at http://dx.doi.org/10.7717/peerj.16727#supplemental-information.

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
