# Peer review of "Factors associated with intention to be vaccinated with the COVID-19 booster dose: a cross-sectional study in Peru"

_PeerJ, doi:10.7717/peerj.16727_

## Round 0.1 · original submission · Major Revisions

The authors should describe the rate of acceptance of the primer doses (first and second doses) in Latin America in the introduction.
Define the abbreviation cited in the manuscript:
Exp: ESAVI, MINSA
In table 1, separate the demographic and health characteristics in table 1 (you can make 2 tables).
In your tables, you can add a column for the demographic and health characteristics (exp: a column for age, sex, residence...and another for the variables: male, female......) to reduce the length of the tables.
You should summarize the discussion and extend the conclusion.

**Language Note:** The review process has identified that the English language must be improved. PeerJ can provide language editing services - please contact us at copyediting@peerj.com for pricing (be sure to provide your manuscript number and title). Alternatively, you should make your own arrangements to improve the language quality and provide details in your response letter. – PeerJ Staff

Reviewer 1 ·

Basic reporting

Abstract
a) Line 36: The word analytic should be removed.
a) Line 99: The words through a primary database should be removed.
Materials and Methods
a) Line 102: The word analytical should be removed.
b) Line 120: It is necessary to clarify who has conducted the interview.
c) Line 219: It is necessary to clarify the meaning of the physical consent for the face-to-face surveys if it was written or oral.
d) It is not stated whether the participant was informed about the use and anonymization of the data and that survey responses guarantee the anonymity of each participant.
e) It is not given any information whether participants were not able to continue to the next question of the questionnaire if they failed to provide a response to an item.
f) It is necessary to clarify whether any sort of follow-up or remind has been conducted.
g) The Authors should clarify about the face-validity testing of the questions with an explanation of the validity of the content of the questions with regard to the research aims. The Authors should clarify how they had estimated the reliability, or internal consistency, of the questions by using, for example the Cronbach’s alpha in order to measure whether or not a score is reliable.
Results
a) Was there any attempt to quantify the response bias: information about non-responders. It would be useful to have some kind of indication of comparability with non-respondents. Is there any population-based data available? How did they differ from those in the sample, how representative is the sample and were the findings representative of Peru?
Discussion
a) The authors should expand their comments regarding the results of previous surveys conducted in different geographic areas regarding the reasons, also in terms of willingness, for receiving the booster vaccination.
b) There is a lack of comparison with the results of recent studies conducted in other geographic areas. The work should therefore be enriched in such a way as to become self-supporting by photographing the context and what is around it. For example, the following articles should be cited Miraglia del Giudice et al. Vaccines 2023;11(4):737; Lee et al. Vaccines (Basel). 2023;11(3):638.
c) The pivotal role of healthcare providers as source of information with a positive impact towards vaccination should be stressed and studies supporting this statement should be added. For example, the following articles should be cited Miraglia del Giudice et al. Vaccines 2022, 10(3), 396; Wang et al. Vaccines (Basel) 2021;9(3):29.

Experimental design

See Basic reporting

Validity of the findings

See Basic reporting

Additional comments

See Basic reporting

·

Basic reporting

NA

Experimental design

NA

Validity of the findings

NA

Additional comments

The manuscript entitled “Factors associated with intention to be vaccinated with the COVID-19 booster dose: A cross-sectional study in Peru” offers a crucial insight into the factors associated with the intention to be vaccinated with the COVID-19 booster dose in Peru. Employing a well-structured analytical cross-sectional design, the research meticulously captures both sociodemographic and clinical factors, providing a comprehensive understanding of vaccine acceptance in the region. Given the global importance of vaccination in combating the pandemic and the unique challenges faced by different regions, this research holds significant value for public health strategies and interventions. I believe the manuscript is suitable to be published in this journal after some minor corrections/improvements.

1) There are some grammatical and typo errors, authors should go through the manuscript and correct them, here are few of those:
1. Intention to be vaccinated (Please see line no. 70).
2. Likewise, physical surveys were applied conducted in public places and no incentive was given to the participants of the study. (Please see line no. 118).
3. These criteria were described by WHO (2l) and have already been used in other research….(see line no. 178).
4. We evaluated the IBV according to the recommendations of various agents. (see line no. 191)
5. To ascertain that she resided in Peru, she was asked in the survey about the current apartment in which she lived. Rephrase this sentence as all the individuals are not female in this study, better use the term “they”.
6. There are instances where consistency could be improved. For example, the term "Intention to be Vaccinated" is sometimes abbreviated as "IBV" and other times as "intention to vaccinate." It would be beneficial to maintain consistency in terminology.

2) The authors should recheck the calculation of the percentage done in the manuscript. In the results and in table sections calculations seem to be incorrect by a small fraction.
1. for example, percentage of females in the study should be 59.67 (or 59.7 if the authors want to represent it in 3 significant figures).
2. The total number of individuals in Age column of Table 1 is not corroborated with the total number of subjects in the study. (614+184+125= 923 not 924)
3. Similarly, the total number of individuals mentioned in the Religion column is also not corroborating with the total number of subjects (individuals) in the study.
4. Also, the column about Education has the same issue.

3) In the discussion section, the outcomes of this study are quite well discussed, however, the authors should try to incorporate and discuss the findings of other similar studies conducted in other regions or countries. They should also mention how other’s results and conclusions corroborate with the findings of this manuscript and how and where the results are contradictory.

---

## Round 0.2 · accepted · Accept

The authors have addressed all of the reviewers' comments.

Reviewer 1 ·

Basic reporting

no comment

Experimental design

no comment

Validity of the findings

no comment

Additional comments

no comment

·

Basic reporting

NA

Experimental design

NA

Validity of the findings

NA

Additional comments

The authors have addressed nearly all the comments and suggestions provided during the review process. The revised manuscript demonstrates a well-structured and meticulously executed analytical cross-sectional study on the factors influencing the intention to receive the COVID-19 booster dose in Peru. The clarity of presentation, including relevant sociodemographic and clinical variables, and robust statistical analyses contribute to the manuscript's strength. Notably, the authors have effectively responded to concerns, enhancing the overall quality of the study. In my opinion, with these revisions, the manuscript is now poised for publication in this journal, offering valuable insights into the factors influencing booster dose acceptance amid the ongoing COVID-19 pandemic.